# Estimation of Late Postmortem Interval: Where Do We Stand? A Literature Review

**DOI:** 10.3390/biology12060783

**Published:** 2023-05-28

**Authors:** Lorenzo Franceschetti, Alberto Amadasi, Valentina Bugelli, Giulia Bolsi, Michael Tsokos

**Affiliations:** 1Istituto di Medicina Legale, Dipartimento di Scienze Biomediche per la Salute, Università degli Studi di Milano, via Luigi Mangiagalli 37, 20133 Milan, Italy; giulia.bolsi@unimi.it; 2Institute of Legal Medicine and Forensic Sciences, Charité-Universitätsmedizin Berlin, Turmstr. 21 (Haus M), 10559 Berlin, Germany; 3South-East Tuscany Local Health Unit, Department of Legal Medicine, via Cimabue 109, 58100 Grosseto, Italy; vale.buge@gmail.com

**Keywords:** postmortem interval estimation, skeletal remains, advanced decomposition, multidisciplinary assessment, late postmortem interval, best practices

## Abstract

**Simple Summary:**

The estimation of the time of death, or post-mortem interval, is a crucial aspect of forensic death-related investigations. However, accurate estimation can be challenging, especially when dealing with cadavers in advanced stages of decomposition and skeletal remains. The decomposition process and taphonomic changes can affect the body’s external appearance, making it difficult to determine even an approximate time since death. Additionally, factors such as temperature, humidity, and the presence of insects and scavengers can accelerate or slow down the decomposition process, further complicating the estimation of postmortem interval. Forensic practitioners rely on various methods, such as the examination of post-mortem changes in the body, the analysis of bone structures, and instrumental and molecular approaches. Despite these efforts and new methodologies, accurate estimation of PMI remains a complex task. The present review illustrates the main methods that can be used in such distinctive cases, seeking to provide forensic experts with useful elements to understand which approach best fits the case they have to evaluate in their ordinary practice.

**Abstract:**

Estimating time since death can be challenging for forensic experts, and is one of the most challenging activities concerning the forensic world. Various methods have been assessed to calculate the postmortem interval on dead bodies in different stages of decomposition and are currently widely used. Nowadays, the only well-recognized dating technique is carbon-14 radioisotope measurement, whereas other methods have been tested throughout the years involving different disciplines with different and sometimes not univocal results. Today, there is no precise and secure method to precisely determine time since death, and late postmortem interval estimation remains one of the most debated topics in forensic pathology. Many proposed methods have shown promising results, and it is desirable that with further studies some of them might become acknowledged techniques to resolve such a difficult and important challenge. The present review aims at presenting studies about the different techniques that have been tested in order to find a valuable method for estimating time since death for skeletal remains. By providing a comprehensive overview, the purpose of this work is to offer readers new perspectives on postmortem interval estimation and to improve current practice in the management of skeletal remains and decomposed bodies.

## 1. Introduction

Estimating the time since death, commonly referred to as the post-mortem interval (PMI), is a crucial task in every death investigation [1,2,3,4]. The longer the time since death, the more imprecise the PMI assessment. Several post-mortem (PM) indicators can be used to establish the time since death in the early PM period. The rigor mortis, algor mortis, and livor mortis are the most common PM changes used in the early PM period by pathologists [1]. Unfortunately, the above PM indicators can only be applied within 48–72 h after death and no longer [1,5]. Additional methods involve supravital reactions of tissue such as electrical stimulation of eyelid muscle, pharmacological excitability of the iris, the analysis of vitreous potassium and hypoxanthine, and the analysis of biological clock genes. However, these methods lose their value and accuracy as decomposition progresses, making it very complex to estimate PMI [4,5].

In everyday activity, finding corpses in an advanced state of decomposition is not uncommon, and presents the forensic practitioner with various challenges, among which estimating PMI is probably one of the most difficult [6]. The situation is even more complex when skeletonised bodies are concerned, which is the case for two reasons: skeletonisation can occur at different times according to various factors (such as environment, soil characteristics, or weather conditions), and it is difficult to distinguish the age of bones by visual observation, as few or no macroscopic changes occur. In forensic cases, circumstantial elements may gain relevance and provide indications about the estimation of PMI; these must be integrated with the scientific laboratory methods we consider later in this review. Acknowledging information about death time can be useful for many reasons, from legislative to investigative, as it can be decisive as to how the investigation is pursued.

Over the years, many techniques have been studied in order to find a suitable and reliable method for assessing PMI from severely decomposed bodies and skeletal remains. In these latter cases, dating skeletonized remains could be of the utmost importance, first of all to differentiate between archaeological and forensic remains, which is fundamental to understanding how to proceed with further investigations. While several of these methods have been studied more widely than others, all of them are in the experimental phase, and research in this field is constantly evolving.

The purpose of this review is to present the different techniques that have been explored until today in order to provide insight into the current research situation by showing which methods have been evaluated and how reliable they are considered to be. Moreover, this paper can represent a useful consultation tool for forensic practitioners who are faced with the challenge of determining PMI from severely decomposed bodies or skeletal remains.

## 2. Methodology

The present systematic review was carried out according to the Preferred Reporting Items for Systematic Review and Meta-Analyses (PRISMA) standards. This method is a widely recognized and comprehensive guideline for conducting and reporting systematic reviews and meta-analyses in scientific literature research. It provides a structured approach to ensure transparency, rigor, and reproducibility in the review process, facilitating evidence-based decision-making [7,8].

### 2.1. Search Criteria and Critical Appraisal

As previously outlined, the purpose of this review is to provide information on the analyses that may be performed on putrefied and skeletal remains in order to establish the late postmortem interval. In doing so, the review defines practical applications for forensic practitioners [9]. This review, to the authors’ best knowledge, is the first to analyze the existing literature specifically about the methodologies that can be applied in estimating the late postmortem interval and its reliability.

To fulfil these objectives, both a systematic literature examination and a critical appraisal of the collected studies were conducted, using PubMed as main database for articles search. In the first phase, keywords were chosen in order to minimize false negatives (e.g., recurring to all possible variants and synonyms for significant terms). The study design included original articles, reviews, case reports, comparative studies, and case series. No unpublished or gray literature was considered.

The following query was used:


*((Postmortem Interval[MeSH Terms] OR Postmortem Interval[Title/Abstract]) OR (PMI[Title/Abstract] AND (skeletal remains[Title/Abstract] OR decomposed bodies[Title/Abstract]))) AND (“Methods”[MeSH Terms] OR “Methods”[Title/Abstract])*


Results were then filtered for publications in English, resulting in 385 publications. This search was last updated in April 2023.

For each paper included in the literature review, the title, authors, journal, year, and type of publication were extracted. Bibliographies of all identified papers were reviewed and compared to identify additional relevant literature. Methodological evaluation of each study was conducted according to PRISMA standards, including assessment of bias. Disagreements on eligibility among researchers were resolved by a consensus process. All researchers independently reviewed papers for which the title or abstract appeared relevant and selected those that analysed postmortem interval with “skeletal remains” or “putrefied bodies”.

In the screening phase, publications clearly falling out of scope with respect to the aim of this review were excluded. After the screening phase, 65 publications were assessed as eligible for full-text assessment. Finally, 22 articles were added through backward search (analyzing the cited references in the selected articles), resulting in 85 articles included in the conceptual review.

Figure 1 shows the PRISMA chart which synthetically describes the screening process.

### 2.2. Risk of Bias

Highlights of this systematic review include the number and breadth of the collected studies, which span the globe; the hand search and scan of reference lists for the identification of all relevant studies; and a flowchart that describes in detail the study selection process. Despite our efforts to fairly evaluate the existing literature, this review includes studies that were published over a time frame of more than 30 years; thus, these results should be interpreted considering that the accuracy of the scientific procedures may have changed over time.

## 3. Results and Discussion

Eighty-six papers dealing with late post-mortem interval and methodological issues that fulfilled the inclusion criteria were included in the investigation. Among these, 58 were original articles, three were case reports, nine were comparative studies, and fifteen were reviews, as reported in Figure 1.

As is well known, in cases of severely decomposed human bodies or remains, bone material is useful for genetic and toxicological study and evaluation [10,11,12]. The skeletal structure is primarily composed of a hard component called the extracellular matrix, which contains a cellular part made of osteocytes. The extracellular matrix is composed of an inorganic part, which is largely comprised of hydroxyapatite crystals, and an organic one containing collagen, proteoglycans, proteins such as albumin and hemoglobin, lipids, and water. As far as the forensic world is concerned, bone degradation is strongly influenced by environmental factors such as soil acidity, the presence of bacteria, and weather conditions. The presented methods analyse bone components with different techniques that are described below under separated subheadings. Various advanced methods that have been proposed and evaluated for late PMI estimation in such cases are discussed in detail.

### 3.1. Radiocarbon (C14) Dating

The most commonly used and well-acknowledged method for assessing PMI is the 14 radiocarbon (14C) level analysis. It is considered a valid technique in several fields, such as archaeology and geology, for dating organic materials including human remains as well as fossils and wood. It is based on the principle that living organisms absorb and store 14C, an isotope of the carbon element, taking it from the atmosphere during their lifetime. However, upon death, this uptake ceases, and 14C begins to decay gradually. Numerous studies have investigated whether the 14C dating method applied to human bones could be suitable for assessing PMI. Ubelaker [13,14,15,16] published a series of studies about radiocarbon analysis in human tissues. One critical aspect to consider is the “bomb curve” phenomenon; after 1950 there was an increase in radiocarbon in the atmosphere, and subsequently in human tissues, due to thermonuclear device testing. This curve reached its peak in 1963. Following this year, the 14C level in the atmosphere started to decrease, but never returned to pre-1950 levels. These artificially high levels of radiocarbon have been captured in organic materials such as human tissues. Therefore, by measuring 14C ratio in bone collagen, it is possible to establish whether the death occurred before or after 1950. After that 14C level has been detected, the following step is to assess whether the values fall on the ascending (1950–1963) or descending (after 1963) portion of the curve. This requires further information, such as knowing the age at death. Studies [15,16] have demonstrated that the relationship between bone formation and 14C content are related, meaning this method could be used to estimate birth time from dental or bone remains. Therefore, by knowing or assessing the age at death, it may be possible to determine the date of death. It is worth noting that bone has a slow turnover rate compared to other tissues, making it less suitable for accurate PMI estimation [17]. Additionally, environmental conditions can significantly impact the 14C content in bone, leading to potential changes in bone composition even in recent remains [18]. Therefore, careful consideration of environmental factors and thorough validation of results are crucial when applying radiocarbon analysis for PMI estimation in skeletal remains.

### 3.2. Not Only C14: Other Radioisotopes

Other radioisotopes have been investigated for their potential use in estimating PMI in human remains in addition to C14. Schrag et al. [19] conducted a study on the behaviour of 90Sr (Strontium) and 210Po (Polonium) in human bones. 90Sr levels were found to follow a similar pattern as 14C, with a “bomb peak” from nuclear testing, making it useful in distinguishing between forensic and archaeological samples. On the other hand, the content of 210Pb showed individual variations, being higher in smokers, underwent diagenetic changes in buried individuals, and was leached away by weather conditions. These individual factors can be taken into consideration when developing a new method for PMI estimation. Another study by Kandlbinder et al. [20] explored the content of radionuclides in bones for assessing PMI. The researchers analysed the relationship between 228Ra (Radium) and its decay product 228Th (Thorium). The amount of 228Ra in bones depended on ingestion, while the transformation rate was independent of external factors. This suggested that measuring 228Th/228Ra ratio could be a starting point for developing a new method for assessing PMI that is not influenced by external factors [21]. However, further research and validation are needed to establish the reliability and accuracy of these methods, taking into consideration factors such as environmental conditions, individual variations, and diagenetic changes.

### 3.3. Fluorescence Techniques: Luminol and Ultraviolet-Induced Light

Other studies have investigated completely different methods based on the observation of the optical behaviour of bones when they are analysed with special techniques involving the formation of a fluorescent light. A number of these are widely validated methods, such as luminol testing and UV-induced fluorescence. Luminol is a chemical compound that, when hit with a special light, produces a blue fluorescence if it is in contact with an activating factor. In forensics, the activating factor is represented by hemoglobin; thus, if skeletal remains show positivity to luminol testing by producing a blue fluorescence, they still contain a certain amount of hemoglobin. The use of this technique over bones to determine PMI was proposed by Introna et al. [22], and has been investigated subsequently by several other authors [23,24,25,26]. The results of these studies have shown that more recent bone remains produce a stronger and wider blue luminescence compared to archeological ones due to hemoglobin loss in the latter. However, se results are not always precise, and there are a certain amount of false negatives [23,24] due to accelerated loss of hemoglobin by external factors. This was demonstrated in a study by Caudullo et al. [24], in which luminol testing was performed twice on the same samples after manipulation and consequent alteration of the bones, with a greater amount of false negative in the second analyses. Lastly, it can be said that analyses with luminol techniques could provide quite precise differentiation between archaeological and forensic bone samples [25,26].

The same principle behind luminol can be applied using other fluorescence-based methods, such as UV-induced fluorescent light [26,27,28,29,30]. This method is based on the observation of the width of the reflecting surface of the bone sample, with a larger area indicating a shorter PMI, as reported by Ramsthaler et al. [26]. In particular, for bones with a PMI of less than 50 years (of forensic interest), the entire surface showed reflectivity. In older samples, the behaviour was not always clear; however, in general when the whole surface shows positivity for this test, the sample is considered to be of forensic interest. Sterzik et al. [27] compared this method to a new one based on 490-nm-induced fluorescence, which turned out to be useful as well for distinguishing between archaeological and forensic remains. Other authors [28,29,30] have explored the use of different colors of UV-induced fluorescence on bones, demonstrating that in recent samples generally return a blue result due to higher presence of collagen, while older samples return a yellow/brown result. A yellow/brown fluorescence is more indicative of a longer PMI, while blue fluorescence is suggestive of a shorter PMI. While this last method is a promising one for estimating PMI, it should be investigated more in order to reduce errors. Fluorescence-based methods such as luminol testing and UV-induced fluorescence show promise as potential techniques for estimating PMI in human remains, and may offer advantages in terms of cost and speed compared to C14 dating.

### 3.4. Bone Extracellular Matrix Component Analyses: Citrate

A different approach has been adopted by several authors who focused their studies on analysing the presence of different bone components, particularly in the bone extracellular matrix. As previously said, as bone undergoes morphological and histochemical changes over time, its composition is modified, with a resulting change in concentration of various elements.

One of those components is citrate. Its content in bones does not vary between men and women, but different bones of the same individual can show different amounts of citrate, and its concentration is lower in osteoporotic bones. [31]. The use of citrate content in bone as a predictor of PMI has shown mixed results in various studies [31,32,33,34]. While certain studies have demonstrated a correlation between citrate content and PMI, other studies have not reached statistically significant results. Schwarcz et al. [31] initially studied the citrate content in bone to determine whether it could be a valid predictor of PMI, demonstrating that it decreased regularly in a series of samples collected in different settings. It appeared that storage conditions, such as temperature, depth of burial, etc., did not greatly influence the rate of citrate loss. The only exception was that there was no loss of citrate at temperatures below 0 °C. Subsequent studies, however, did not reach the same results. Wilson et al. [32] tried to replicate the research by Schwarcz et al. Although they did not reach the same results, they observed a correlation between citrate content and PMI. The reason why they did not reach statistically significant results may be that the bone samples were kept in an artificial and aseptic environment. In fact, the previous study had suggested that water could be necessary to observe the postmortem changes in bone citrate content; in the later study, with the environment involving sealed containers and distilled water, the changes may not have been the same. Another reason may be that the bone samples used in this study were too young; they used pig bone samples, whose composition is analogous to that of a five-old human, and the initial citrate content may have been too low to reach satisfying results. Other factors, such as storage conditions and burial environment, may influence the degradation rate of citrate in bone as well. Kanz et al. [33] observed that the degradation rate of citrate seemed to be slower in soil-buried bones compared to those deposited on the surface. Brown et al. [34] did not find significant correlation between citrate content and PMI, although they did observe a decrease in citrate content with increasing PMI.

While citrate content in bone has been proposed as a potential method for estimating PMI, the results from studies have been inconsistent. Further research is needed to better understand the factors that may affect citrate content in bone and its reliability as a predictor of PMI.

### 3.5. Proteomics

Proteomics, or the study of proteins, has been used as a method to analyse bone samples for estimating PMI [35,36,37,38,39,40,41,42,43,44,45,46]. Several studies focused on early PMI, obtaining remarkable results [41,42,43,44,45,46]. Other issues arose with respect to late PMI. Procopio et al. conducted a study using proteomics to investigate the degradation patterns of bone proteins over time in pig carcasses with known PMIs [36]. Proteins contained in bone can degrade in different ways, such as deamidation or hydrolysis. Analyzing pig bone samples, the study provided an overview of bone proteins and their transformation/degradation through time. The study identified four groups of proteins with distinct degradation patterns in relation to PMI. Additionally, the concentration of the protein fetuin-A was found to have a negative correlation with biological age at the time of death, which was consistent with a previous study [37]. Conversely other serum proteins, such as alpha-1-antitrypsin and chromogranin-A, showed an opposite trend, with their concentration increasing with biological age. Another study by Costa et al. [38] demonstrated that biglycan deamidation levels were statistically different as a function of PMI. In their study, transferrin decreased with a half-life of one month at room temperature (21 °C); however, in the study by Procopio et al., the burial temperatures were lower, and transferrin could be found after six months postmortem, indicating that environmental factors such as temperature can impact protein degradation in bone. Furthermore, Prieto-Bonete et al. [39] proposed a profile of 32 proteins that could discriminate between bones with a PMI lower or higher than twelve years, suggesting that proteomic analysis of bone proteins could be a useful method for assessing PMI in recent cases. In a study by Bonicelli et al., liquid chromatography mass spectrometry was used to obtain untargeted metabolomic, lipidomic, and proteomic profiles in bones. The combination of these biomolecules’ classes in a multi-omics model could therefore be beneficial for estimating PMI across a broader range of potential PMIs. Metabolites and lipids offer accuracy in the short to medium term, while proteins could be the main markers for longer PMIs due to their greater stability [46].

### 3.6. Small Particles for PMI: Evaluation of Bone Components including MicroRNA, DNA Concentration, and Collagen Degradation

Several studies have focused on investigating different biomarkers in bone samples. Joo-Young Na conducted a study in 2018 [47] on the usefulness of microRNA in bones for evaluating PMI up to two years. A negative correlation between target microRNA expression and PMI was observed. Bone samples were divided into four groups based on PMI, and it was found that the expression of target microRNA in bone samples from group 1 (PMI < 1 month) was statistically different from other groups. Other research focused on microRNA with similar conclusions and shorter PMI [48,49,50].

Perez-Martinez et al. [51] assessed the usefulness of various biochemical parameters to estimate PMI in a study involving 80 long bones with PMI ranging from 5 to 47 years. The samples did not come in contact with the ground during burial, minimizing the process of transformation and diagenesis. The bones were divided into two groups: PMI lower than 20 years, and those more than and including 20 years. The study showed a significant decrease in nitrogenous bases (adenine, guanine, purines, cytosine, thymine, pyrimidines) and peptides in collagen type I in bones with PMI ≥ 20 years. However, no correlation was found between DNA concentration and increase of PMI [52]. Instead, an inverse correlation between DNA fragment length and PMI was found, suggesting that further investigation of this relationship could lead to the construction of a new valid method for assessing time since death in bone remains. Hagelberg et al. [53] demonstrated that DNA degradation in bones was not dependent on PMI, but rather affected by bone preservation, which was in turn conditioned by environmental features such as soil composition.

Other studies evaluated collagen degradation as a potential biomarker. Boaks et al. [54] conducted a study on porcine bones with PMI up to twelve months, using histochemical staining to distinguish between collagenous (Co) and non-collagenous (NonCo) proteins and then analysing the ratio between them using spectrophotometry. The study revealed a significant decrease in the Co/NonCo proteins ratio with an increase in PMI. Jellinghaus et al. [55] attempted to reproduce the same method on human bones, however, their findings were less significant; thus, they applied a different method to detect the Co/NonCo protein ratio, demonstrating that a significant reduction in it is correlated with PMI for male individuals, confirming that it could be an interesting factor for narrowing down the PMI interval.

### 3.7. Radiology and Spectroscopy

X-Ray diffraction (XRD) can be a useful mean to determine the crystalline structure of bones, which undergo changes after death similar to other organic matter. A study conducted by Prieto-Castellò et al. [56] analysed bones with a PMI ranging from 7 to 54 years. A cortical and medullar sample from each fragment of bone was subjected to XRD analysis; moreover, an evaluation of organic matter was made using different methods, each suitable for detecting different components. The burial conditions were kept consistent for all the samples, with minimal interaction with the environment, to reduce external influences on degradation. The analysis of organic component concentration modifications revealed interesting results. There was a statistically significant correlation between sulphur concentration and PMI, indicating that phosphorus concentration increased with time since death. In addition, there was an increase in phosphorus concentration with PMI, suggesting that phosphorus concentration changed with time as well. On the other hand, urea concentration showed an inverse relationship with time since death, indicating that urea concentration decreases with increasing PMI.

Another potential approach for characterizing bone components at the molecular level is Raman spectroscopy [56,57,58,59,60,61,62,63]. Through the emission of electromagnetic radiation, it can detect the presence of collagen and mineral phases that constitute the extracellular matrix [57,58,59]. The objective of this study was to evaluate the suitability of Raman spectroscopy for determining the PMI by analyzing different bone elements based on previous research conducted by McLaughlin and Lednev [59] on trends in the chemical composition of buried bones. It is noteworthy that the diagenesis of bone can be influenced by soil bacteria, particularly bacterial collagenase activity that degrades bone collagen. A previous investigation [60] on teeth from archeological contexts demonstrated a correlation between burial duration and PMI. Building on this, McLaughlin and Lednev hypothesized that it may be feasible to develop a model based on Raman spectroscopy results and known diagenetic patterns to assess burial duration. The authors concluded that Raman spectroscopy was a valid method for detecting chemical changes in bones with a short burial duration, as they found a high correlation between the two. In a similar study, Creagh and Cameron [61] conducted an analysis of unburied animal bones using Raman and infrared spectroscopy based on the CH–aliphatic ratio, which is an organic-to-mineral ratio used as an indicator of the amount of lipids and other organic materials in bone. They observed that the decrease in CH–aliphatic ratio with increasing PMI was species-dependent, suggesting the need for further investigations using human remains to determine if this could be a reliable method for PMI estimation.

Ortiz-Herrero et al. [62] conducted a study on 53 human skeletal remains with PMIs ranging from 15 to 87 years, focusing on the analysis of the cortical area of the bone, which contains both organic and inorganic components such as hydroxyapatite crystals that contribute to bone hardness. As previously mentioned, bone degradation is influenced by various internal and external factors, resulting in the loss of collagen, nitrogen, and amino acids. In this study, Raman spectroscopy was employed together with a multivariate calibration technique as a non-destructive, versatile, and accurate method for analyzing bone composition. The objective was to examine the modifications in bones with known PMIs. The PMIs of 10 out of 14 validation samples were determined with an accuracy error of less than 30%, revealing that Raman spectroscopy holds promise as a method for PMI estimation.

Another technique commonly used for bone composition analysis is Fourier transformation infrared (FT-IR) spectroscopy [64,65,66,67,68,69,70,71,72,73]. In a study conducted by Nagy et al. [64], FT-IR spectroscopy was employed to differentiate between archaeological and forensic bone remains. The forensic samples belonged to skeletons that had died in 1980, while the archeological samples belonged to skeletons that had died before 1500. The researchers identified francolite, an inorganic component, which was only present in archaeological bones and not in forensic ones. This suggested that francolite could serve as a useful marker for distinguishing between archeological and forensic bones when estimating PMI. Furthermore, the study found that archaeological samples had an overall higher representation of inorganic components, while collagen and other organic elements were more prevalent in forensic samples. Differences in crystallinity index and carbonate/phosphate (C/P) ratio were detected as well, which were consistent with differences observed between healthy and unhealthy bones. Similar results were obtained by Patonai et al. [65], who showed that forensic bones contained more organic elements and exhibited a complex pattern of changes in bone composition with increasing PMI. In addition, changes in crystallinity index were observed, indicating alterations in the crystal structure of apatite due to diagenesis. Wang et al. [66] attempted to combine FT-IR spectroscopy with chemometrics to develop a model for dating human remains. They analysed buried and unburied bones and found that degradation was considerably slower in unburied ones, likely due to the unfavorable conditions for microbial growth and mineralization in ventilated and dry environments. This had been previously seen by Howes et al. [67], who observed greater changes in organic and carbonate components and crystallinity index when examining bones buried in an acidic soil. Although a specific model based on FT-IR spectroscopy and chemometrics was not created in their study, the findings of Wang et al. findings showed promising results for future research, highlighting the need to consider environmental factors when investigating these methods.

Longato et al. [74] compared three different techniques for bone component analysis, namely, micro-computed tomography (micro-CT), mid-infrared (MIR) microscopic imaging, and energy dispersive X-Ray (EDS) mapping. Micro-CT, which allows visualization of the internal structure of bones, was useful in demonstrating that the more recent samples had a higher bone density [75]. On the other hand, MIR microscopy provided a view of the bone surface and allowed the creation of a “bone map” indicating the locations of organic and inorganic components. In particular, archaeological bone samples were found to have higher fluoridation compared to forensic samples through this method. EDS mapping, which shows the distribution pattern of elements, was used to describe the decomposition of bones and determine calcium-to-phosphorus (Ca/P) and calcium-to-carbon (Ca/C) ratios. These indexes varied in bone composition with time since death, with the Ca/P ratio increasing as PMI progresses and the Ca/C ratio specifically increasing with degradation of organic matter and increasing mineralization.

Le Garff et al. [76] conducted a study using bones with a short PMI. They analysed bone samples using micro-CT, observing trabecular modifications through three-dimensional reconstruction of images. Ten bone samples were used for this study; they were collected shortly after death and observed through micro-CT every two weeks for ten weeks. The results were significant for the first two weeks, with an increase in trabeculae separation and a decrease in body surface/body volume ratio during this period. After two weeks, no statistically significant changes were found.

In a recent study, Schmidt et al. [77,78] divided 99 bone samples into five classes according to their PMI (which ranged from 1 day to 2000 years) and then evaluated them with near-infrared (NIR) spectrometry, which provides a quantitative analysis of different physico-chemical parameters as well as other techniques that have previously been tested in other studies. The results showed significant differences in bone samples between classes 1–4 (with a maximum PMI of ten years in class 4) and class 5 (PMI > 100 years), indicating that NIR spectrometry can be useful in distinguishing between forensic and archaeological bone remains.

### 3.8. Taphonomic Analysis and Morphological Assessment

It is important to note that estimating PMI from bodies that have undergone major changes after death can be extremely difficult. The first to evaluate such changes from a quantitative point of view were Megyesi et al. [79]. In their study, Megyesi et al. developed a model for estimating time since death based on a quantitative assessment of decomposition using a total body score (TBS) method. TBS is based on four categories of decomposition (fresh, early decomposition, advanced decomposition, and skeletonization) [80], which are further subdivided into point-valued stages describing the morphological appearance of postmortem changes in different anatomical regions. The scores assigned to each region are combined to produce a TBS ranging from 3 (fresh in all regions, with no discoloration) to 35 (dry bone in all regions). This model has been used in several other studies by different researchers [4,12,81,82,83,84,85,86,87,88,89,90,91,92,93,94,95,96,97,98,99].

Using this objective quantified scale of decomposition, Megyesi et al. [79] were able to highlight the fundamental relationship between decomposition and time/temperature (ADD). ADDs (Accumulated Degree Days) is a measure of heat energy units required for a biological process or organism to develop from one life stage to the next. Simmons et al. explained this concept as ADDs representing the energy that is placed into a system through the accumulation of temperature over time. ADDs can incorporate time and heat in a single value and can be easily determined by experimentation [92]. Megyesi et al. proposed a model improved by Moffat et al. [93], while Heaton et al. [94] proposed a modified one for bodies in water. The result of these equations provides the number of ADDs needed for a cadaver to reach a given TBS assessed by the forensic pathologist or anthropologist [4]. 

The efficacy of Megyesi’s formula in different environments and under different conditions remains under evaluation and discussion in the scientific community; a number of studies have been conducted, reaching different and controversial results [4,5,83,84,88,95,96,97,98,99]. Further studies are needed for the validation of the method, as precision is affected due to intrinsic biological and environmental variability.

Worth mentioning are the several outdoor facilities operations today for the study of human decomposition [85,86,96]. These places, known as body farms or decomposition facilities, play a crucial role in PMI estimation studies. These unique environments offer researchers the opportunity to study the decomposition process of human remains in a controlled yet natural setting, thereby generating invaluable data for the understanding of various factors affecting decomposition rates. For instance, Suckling et al. [86] on human outdoor decomposition effectively illustrated the influence of factors such as climate, insect activity, scavenger behavior, and soil composition, which are incredibly relevant in estimating the time since death. This understanding is vital in forensic science, making outdoor facilities a cornerstone of PMI research and significantly contributing to advancements in forensic taphonomy.

### 3.9. Entomological Approach

Insects are the most numerous and diverse organisms on Earth, found in almost all the terrestrial habitats and in most aquatic ones as well, excluding seawater. While an estimated 3 to 30 million species [100], insects have evolved wings, a feature that distinguishes them from all other invertebrates, which enable them to travel long distances in search of food and suitable egg-laying sites. Arthropods, including insects, are able to quickly recognize the breakdown products of muscle and fat through volatile organic compounds, which vary during decomposition, and use olfaction as their primary long-distance tool to detect, locate, and utilize carrion, making them highly relevant in forensic investigations, including PMI estimation [101,102,103,104,105,106,107]. According to the best practice in forensic entomology [108], during the early PM period, the age of the oldest individual arthropods that have developed on the body can provide crucial information for estimating the PMI. During the late PM period, the composition of the arthropod community in relation to expected successional patterns is used to estimate the PMI. The study of arthropod succession allows experts to associate different species or groups with specific decomposition stages. In this review, the focus is on the late PM period. As soft tissues decompose, they undergo a series of changes that provide specialized resources for different species [109]. Numerous succession studies have been carried out [110,111,112,113,114,115,116] to understand the sequence in which species are attracted to the different stages of decomposition and to estimate the PMI by successional changes over decomposition [117]. Information provided by succession studies has disclosed the potential use of insect community succession in forensic cases. However, succession studies are highly dependent on the habitat and microclimate, and can be affected by several factors. Nevertheless, it is important to know the local fauna in order to evaluate the entomological findings in each single case [118]. 

Two major taxa of insects are predictably attracted to cadavers and provide the majority of information useful for forensic investigations: blow flies, mainly Callipohoridae (order Diptera), and beetles (order Coleoptera). 

Blow flies are of particular forensic interest due to their preference for protein-rich substrates, such as decaying soft tissues, where their larvae can complete their development [109,119,120,121]. Female blow flies are attracted to ammonia-rich compounds and hydrogen, and they often deposit their eggs or maggots deep into openings of the cadaver, including open orifices and wounds, within hours or soon after death. Oviposition does not occur in dehydrated or mummified tissues, as the eggs and larvae require a moist environment for successful development [121]. In arthropod succession, Coleoptera usually follow blow flies. Carrion beetles are attracted to volatile organic compounds (VOCs) emitted by carcass tissues or other insects present on a corpse [122]. 

Physiological experiments have revealed a close relationship between particular species of beetles and VOCs. The production of VOCs attracts carrion beetles, and their appearance on the cadaver is closely related to temperature [123]. Studies by Matsuweski et al. [124,125,126,127] have demonstrated that the time of appearance of carrion beetles on a carcass is influenced by factors such as ambient temperature, onset of bloating, and seasonal, annual, and environmental differences in temperature. In cases of severely decomposed bodies or skeletal remains, insect colonization can be used to assess any disturbances or movements of the remains after death. The colonization patterns and insect species can provide clues about the location and conditions of death, such as whether the body was exposed to the elements or buried. Additionally, the presence of specific species can indicate the presence of a buried or concealed skeleton. Even if forensic entomology does not provide a precise PMI estimate, it can help to provide important insights into cases involving human remains [6,106,128,129].

### 3.10. Comparative Studies

Studies have performed comparisons among the most studied techniques in cases of PMI estimation in skeletal remains. Amadasi et al. [130] conducted a study of 24 bone samples with a known PMI, analysing them with different physico-chemical methods such as FT-IR, EDX analysis, and the XRD method. FT-IR was found to be suitable for detecting collagen composition in bones, with collagen being more represented in recent bones. EDX spectroscopy was useful for assessing the C/P ratio, which was found to be much higher in forensic samples. XRD was able to distinguish between “wet” bone (containing large amount of collagen) and “dry” bone. The presence or absence of collagen determined differences in crystalline grain size, which could serve as a threshold between archeological and forensic samples. Cappella et al. [131] compared three methods of evaluation of PMI on bones: 14C analyses, luminol testing, and evaluation of histological features through the Oxford Histology Index (OHI). The study analysed 20 skeletal remains. According to the 14C analysis method, 2 out of 20 cases were determined to be of forensic interest. OHI scores of 0 (indicating no original histological bone features) were observed in 8 out of 20 cases, suggesting a PMI of over 200 years. OHI scores between 4 and 5 were observed in 9 out of 20 cases, with two of these being the cases identified as of forensic interest using the 14C analysis method. Luminol testing showed strong chemiluminescence when applied to human blood, but no chemiluminescence in the negative control (chalk powder). The two cases of forensic interest showed weak chemiluminescence. The conclusion of this study was that while 14C remains the gold standard procedure, screening involving histological features analysis and luminol testing could be useful in discriminating between archaeological and forensic bone samples.

## 4. Conclusions

From this review, it is evident that estimating the PMI in advanced decomposed bodies or in skeletal remains is a challenging task that requires the use of multiple methodologies. The process of decomposition is complex and is influenced by various factors, such as environmental conditions, temperature, humidity, and presence of fauna, which can significantly affect the estimation of PMI. 

What emerges from this research is that one of the primary challenges in estimating PMI in severely decomposed bodies is the lack of reliable and consistent markers of time since death. Moreover, the loss of soft tissue and destruction of anatomical structures, along with the influence of environmental factors and scavengers, leave only the skeletal elements available for further analyses. In this case, the estimation of PMI become more complex and prone to inaccuracies.

The complex nature of decomposition processes necessitates a variety of techniques, such as the ones highlighted above, that forensic practitioners should know and apply in order to obtain a more accurate and reliable estimation of the time since death. It is crucial to recognize the significance of utilizing a multidisciplinary approach in death-related investigations involving severe decomposition, as decomposition rates and patterns can vary greatly depending on environmental factors, individual factors, and other well-known variables, and relying on a single method may result in inaccurate estimations. 

The choice of methods will depend on the specific circumstances of each case and the available resources and expertise of forensic practitioners. A variety of methods have been used in practice to determine forensic significance, with 14C being the gold standard for use. The most common methods utilized are taphonomic, morphological, and entomological assessments, and there is growing research and interest in the fluorescence techniques (e.g., luminol, UV light), bone matrix component analysis (e.g., citrate, proteomics), small particle analysis (e.g., DNA and collagen degradation), radiology, and spectroscopy. According to best practice, in PMI estimation the available methods are numerous, and their reliability varies, as not all these methods are fully developed and/or validated [132]. The forensic practitioner must take this into account and not rely on a single method of evaluation, and must keep in mind the limitations of the technique to be applied to the specific case.

The fields of application of these methods can be separated into two large groups: techniques for assessing whether the discovered remains are of archaeological interest or forensic interest. From a practical perspective, in cases of discovery of human remains it is important to establish whether they are recent or archaeological in order to determine whether a crime may have been committed. Then, when forensic practitioners are sure that the remains are of forensic interest, they must apply more easily achievable methodologies that can be performed during postmortem examination, such as morphological assessment, together with the application of radiological and spectroscopic methods. In addition, one can make use of more advanced methodology of analyses involving the molecular composition of bone, such as proteomics, DNA, or microRNA, as illustrated in Figure 2. Alongside this, it must always be considered how the surrounding environment can provide elements that can influence PMI estimation.

Further research and advancements in forensic science are necessary to improve the precision and validity of PMI in such peculiar, though not so rare, events in ordinary forensic practice. Therefore, to avoid misinterpretations and evaluation errors in discrimination between bodies of forensic or archaeological interest, correct training in forensic pathology should include expertise regarding the appropriate methodology to be applied in these cases. The use of appropriate methods can provide valuable insights into the time of death. These methods should be based on sound scientific principles validated through rigorous testing and constantly updated with the latest advancements in technology and scientific knowledge. By investing in research and development of specifically validated methodologies, the ability to estimate late PMI in such cases could be improved and thereby contribute to the advancement of forensic science. Close co-operation is highly encouraged in order to provide high-quality death investigation and more accurate and precise evaluation.

## Figures and Tables

**Figure 1 biology-12-00783-f001:**
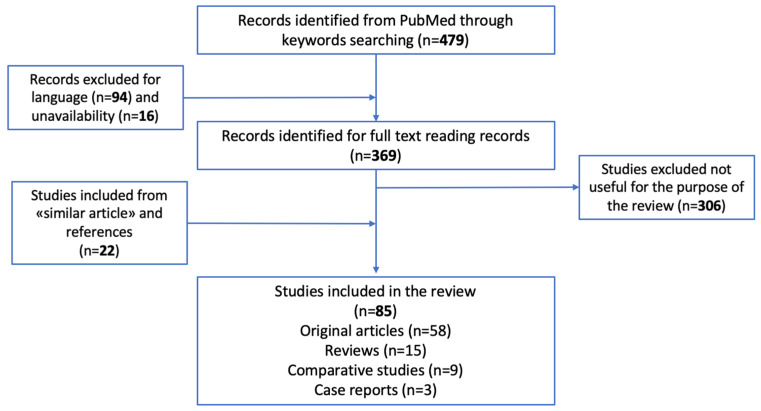
PRISMA review chart.

**Figure 2 biology-12-00783-f002:**
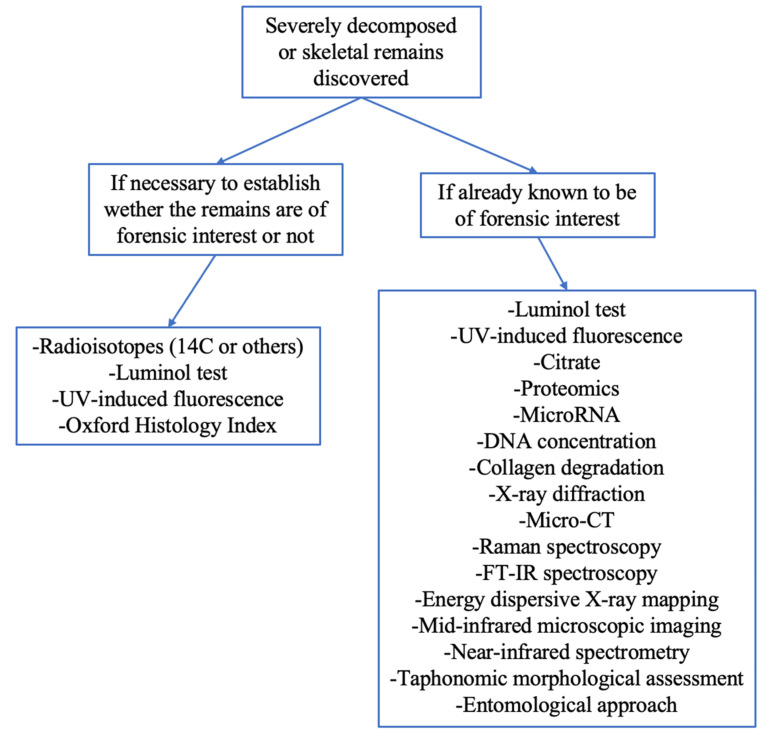
Illustration of the flow chart to be followed in case of late PMI estimation.

## Data Availability

Data are available upon request from the authors.

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
