# Peer review of "Estimation of Late Postmortem Interval: Where Do We Stand? A Literature Review"

_biology, 2023, doi:10.3390/biology12060783_

Round 1

Reviewer 1 Report

The manuscript provides a summary of various methods used to estimate the PMI for remains in the stage of advanced decomposition. The title indicates the methods cover late PMI estimation, which appears to mean skeletonized or mostly skeletonized remains. In my experience in the US, TSD or PMI is often not performed for casework due to issues surrounding the validity and accuracy of the estimations. Regardless, the list of methods provides an excellent collection of data for researchers. I wish the Pub search would have included TSD to be more inclusive.

There should be some distinction in how these methods are used in practice. For example, C14 dating is typically used to determine medicolegal significance vs historic/prehistoric (or contemporary vs non-contemporary); while entomological evidence is to establish a timeframe for more recent remains.  So, while C14 is validated; it is often the less useful method for casework. In fact, contextual analysis and interpreting the taphonomy is often sufficient.

These methods should be mentioned in the summary as important aspects of PMI estimation and interpretation. However, it is understood that these are more subjective approaches. Often, strong law enforcement and medicolegal death investigation work is sufficient to establish a reliable TSD/PMI. While the manuscript is focused on more laboratory or instrument-based methods, investigation and context interpretation should not be ignored.

A pile of newspapers outside a house can often provide a better estimate than current methods. The authors may also want to mention the numerous outdoor research facilities that are in operation today. These facilities contain a plethora of data for PMI studies. However, even with the large number a donors processed at these facilities we have not seen large advancements in PMI estimation methods.

Overall, this is a thorough review of scientific methods for PMI estimation. One final recommendation is to remove any mention of Arpad Vass. Vass is a highly controversial “researcher” who lives in the realm of junk science. He promotes witching or dowsing to find remains and is not well respected in the scientific community.

Reviewer 2 Report

Line 29 – “The present research” should be changed because this is not a research article. This article represents a literature review.

Line 30 – I don’t think “peculiar” is the best word to use

Line 32 – “Estimating time since death from dead bodies” is redundant. You can just say “Estimating time since death can be challenging for forensic experts.”

Line 34 – change “different grades of preservation” to “different stages of decomposition”

Line 35 – Validated through what organization or institution?

Line 40 – I sm confused what is meant by validated. Many forensic practitioners use techniques that have been published (tested with a large data set) but not necessarily validated.

Line 43 – should say “for” skeletal remains.

Line 45 – you reference many studies that include soft tissue assessments, so this is not just about skeletal remains

Line 51 – Remove “is” before “time since death”

Line 68 – This is not a full paragraph

Line 88 – should say “on” putrefied and skeletal remains

Line 89 – change “implications” to “applications”

Line 103 – “filters” should be changed to “filtered”

Line 136 – What do you mean by matrices? Are you referring to the bone material? This is an awkward and long sentence.

Line 142 – This sentence is strangely worded as well. I think you are trying to say “Bone degradation is strongly influenced by environmental factors such as ...”  Therefore, skeletal composition is correlated more strongly with the environment in which the body decomposed than with absolute time since death.

Line 149 – As someone with a long history of analyzing skeletal remains in forensic contexts, I can say that carbon 14 is not the most common or widely accepted technique for modern forensic cases. I’d also like to see some clarification on what is meant by “validated”.  But yes, it is commonly used in archaeology.

Line 183 – “was leached away by weather conditions” ß can you explain this more for readers that are not familiar with isotope analysis.

Line 185 – change “estimate” to “estimation”

Line 204 – change “bone remains” to “skeletal remains”

Line 216 – remove “the” from in front of PMI

Line 237 – remove “represented by”

Line 239 – should say “amounts of citrate, and its concentration is lower in osteoporotic individuals.”

Line 274 – since this is just a review paper and not a research paper it would be beneficial to readers to better define terms and concepts; for example, deamidation and hydrolysis

Line 344 – “archeological teeth remains” should read “teeth from archeological contexts”

Line 423 – remove the word “to” before “were”

Line 492 – remove “have shown a close relationship between particular species of beetles and VOCs”

Line 529 – This is not a complete paragraph

Line 532 – This is not a complete paragraph

Line 538 – change “make usually bones the only matrices” to “leave only the skeletal elements”

Line 539 – should read “for” further analysis

This paper would benefit from being read and edited by an English speaker. I made a few suggestions (see previous section), but I do not have the time to do extensive edits on the grammar and word usage. 

Reviewer 3 Report

The estimation of the postmortem interval is a vital aspect to forensic investigations, but is hampered by the lack of validated methods for accurate and precise predications.  This is especially true for the late postmortem interval. This review manuscript provides a timely review of the status of the science as it relates to estimating the postmortem interval and includes many references that are not in the commonly used forensic sciences journals and, thus, are often overlooked by students and researchers.  Through the use of the PRISMA standards, the authors were able to critically evaluate the relevance of research to provide a more refined scope of this review.  As the manuscript highlights, there are a variety of methods that are being used in practice to determine forensic significance with 14C being one that has been fully-validated for use. When significance is known, there are several options available but reliability of these methods varies, as not all of these methods are fully developed and/or validated, and more work is needed in these areas.  The most common methods utilized are taphonomic morphological and entomological assessments, but there is growing research and interest in the fluorescence techniques (e.g., luminol, UV light), bone matrix component analysis (e.g., citrate, proteomics), small particles (e.g., DNA and collagen degradation), and radiology and spectroscopy. Over the last several years, advances in PMI estimation research have been made, so now is the time to understand the need for validating and refining methods to be in line with best practices.

Overall, I found this manuscript to be well developed and scientifically grounded.  It conveyed many high-level processes in a manner that non-specialists can understand, which makes the paper more accessible to readers.  I do not have any concerns with content, as the thoroughness of the review is one of its strengths.

There are several manuscript questions or concerns, I would like you to consider

11.     One of your key words is best practice, but is not really elaborated upon except for a few passing references to it.  Specifically, in last paragraph of manuscript (starting on line 569) it would be beneficial to the reader to know what is considered best practice for PMI methods. Please consider an additional sentence to more clearly stress this important point.

22.     In methods, lines 84-85, you cite that the PRISMA standard were used. Consider providing a brief summary of what this beyond citing the references 7-8.

33.   Review manuscript for consistent reference to research. In some section/paragraphs the author(s) and year are noted but in others they are not.  This recommendation is for the sentences where you call out a specific research group. For example lines 367-368, but you do not do the same in 412-413.

44.    It is not clear in the beginning of the manuscript how you are defining forensic bone (line 379) vice modern sample (line 399).  I know you are using the language from the references cited, but I think it may be important to the reader to make it clear that ancient and archaeological are equivalent and modern and forensic are equivalent. The term modern is a difficult term to contextualize when research being reported on spans several decades.

54.     I did not see any citations relating to Bonicelli et al. research.  This group has completed several proteomic based studies. One I thought may be of interest to you is doi.org/10.7554/eLife.83658

There are several minor editorial considerations

1.       Line 20- recommend adding (PMI) after interval since the abbreviation is not introduced prior to use.

2.       Lines 68-70- This sentence is out of place for a paragraph on its own. Recommend moving to line 67 and the previous paragraph is recommended

3.       Line 103- filters should be filtered

4.       Line 125- describe should be describes

5.       Sentence lines 149-150- word missing, suggest “only validated method to date” instead of “only validated to date”

6.       Sentence lines 184-185- Unclear what “these features” are referring to. Recommend “these individual factors.”  Factors is what is used in the rest of the paragraph.

7.       Sentence 219-221- distinguish should be distinguishing

8.       Line 216- remove “the” before PMI

9.       Sentence 237-239- missing word, recommend osteoporotic bone instead of osteoporotic

1   Sentence 242-244- I believe “showed” should be “studied” based on the sentence

1   Sentences in 259-260 and 260-261- repeat phrasing “degradation rate…surface”

1   References to Procopio research as cited should be Procopio et al. in lines 271 and 284

1  Line 290- remove extra “small” to read “small particle for PMI”

1  Line 312- evaluate should be evaluated

   Sentences 316-318 and 318 to 320. Very confusing on findings.  In first sentence you state Jellinghaus et al attempted to reproduce the same method but in the second sentence you state they used a different method.  Please rephrase for clarity as what you are referring to in their research.

1   In sentence 329-332, colon was used but a list was not provided as would be expected. Recommend converting to period and staring new sentence with “There was..”

1  In sentence 342-343- what does their in “their collagenase”  refer. Revise sentence for clarity

1  Line 355, last word estimate should be estimation

1 Paragraph 356-365- This paragraph should be revised. There are a lot of repeat phrasing that detracts from overall point. In particular, it is confusing what the Ortiz-Herrero research found since the last sentence of the paragraph cites a different study that wasn’t discussed. I understand what the last sentence was, but it is unclear the significance of this particular work.

2Line 405- “over bones”  suggest replacing over with “using” or something similar

2 Line 415- missing word “in”- should be “differences in bone”

2 Line 462-  PM abbreviation is not previously defined, recommend spelling out in first use. Also, recommend clarifying “individuals” to “individual arthropods”

2Line 473- remove “every” in “every succession”

2 Lines 488-490 should be moved up with prior paragraph on line 487

2  Lines 491-492- repeat phrase should be deleted “close relationship…VOCs”

2 Line 506- revise for clarity of intent.

2Lines 532-534 should be joined to prior paragraph in line 531

2Sentence lines 537-539 unclear as written.

2 Figure 2- first box, I believe “discovering” should be “discovered”

3Figure 2- second line, box on left “wheter” should be “whether”

3 Recommend merging sentence in line 566-568 with final paragraph starting on line 569

3 Line 568- switch forensic and ordinary placement

Overall, the quality of English language is good. 

However, there are several instances where present tense is used but past tense should have been used. Please proof-read manuscript for tense shifting between present and past tense.

Reviewer 4 Report

This paper is a meta-analysis of methods for estimating the date of death of an individual (late postmortem interval) represented by highly degraded and often exclusively skeletal remains, in order to determine whether it is a case of archaeological or forensic interest.
This meta-analysis is well conducted and follows PRISMA standards.  
At the end of the study, the authors propose a decision tree recommending the methods to be used to establish the PMI according to the context.
The article is well written, correctly structured and constitutes a synthesis shedding light on the difficult question of estimating the PMI in the face of skeletal remains.
